# Tracing the invasion: Wing morphometrics reveal population spread and adaptation patterns of *Halyomorpha halys* (Stål, 1855) across Southern Europe

Martina Pajač Beus[1], Darija Lemic[1,2]*, Hugo A. Benítez[2,3,4,5]*, Laura M. Pérez[6], Mojca Rot[7,8], Aleksandra Konjević[9], Stefanos S. Andreadis[10], Ivana Pajač Živković[1]

**1** Department for Agricultural Zoology, University of Zagreb, Faculty of Agriculture, Zagreb, Croatia, **2** Research Ring in Pest Insects and Climate Change (PIC2), Santiago, Chile, **3** Laboratorio de Ecología y Morfometría Evolutiva, Centro de Investigación de Estudios Avanzados del Maule, Universidad Católica del Maule, Talca, Chile, **4** Instituto One Health, Facultad de Ciencias de la Vida, Universidad Andrés Bello, República 440, Santiago, Chile, **5** Cape Horn International Center (CHIC), Centro Universitario Cabo de Hornos, universidad de Magallanes, Puerto Williams, Chile, **6** Departamento de Ingeniería Industrial y de Sistemas, Universidad de Tarapacá, Arica, Chile, **7** Institute of Agriculture and Forestry Nova Gorica, Nova Gorica, Slovenia, **8** Department of Agronomy, Biotechnical Faculty, University of Ljubljana, Ljubljana, Slovenia, **9** Center of Excellence, Faculty of Agriculture, University of Novi Sad, Novi Sad, Serbia, **10** Institute of Plant Breeding and Genetic Resources, Hellenic Agricultural Organization-Dimitra, Thermi, Greece

* dlemic@agr.hr (DL); hbenitez@ucm.cl (HAB)

**Data availability statement:** All data used in this study are publicly available and accessible at: 10.5281/zenodo.14632099 and https://zenodo.org/records/14632099.

**Funding:** This study was supported by Agencia Nacional de Investigacion y Desarrollo in the form of a grant awarded to HAB ANID/ANILLO/

## Abstract

Invasive species such as *Halyomorpha halys* (Stål, 1855), the brown marmorated stink bug, pose a significant threat to agriculture due to their rapid spread and adaptability. The aim of this study is to assess the phenotypic variability of *H. halys* populations by analyzing the size and shape morphology of the anterior and posterior wings and to determine whether geometric morphometrics can serve as a cost-effective alternative to genetic methods for tracking invasion patterns. Populations from four southern European countries (Croatia, Serbia, Slovenia and Greece) with 540 specimens and 2,160 wings were analysed and showed clear phenotypic differences in wing morphology consistent with the known invasion dynamics previously determined by genetic studies. Mahalanobis distances highlight the close morphological relationship between the Serbian and Slovenian populations, suggesting common ancestry or recent gene flow, while the Greek and Croatian populations show significant differences, suggesting different invasion pathways or rapid morphological adaptation. The Greek population exhibited lower phenotypic plasticity, whereas the Serbian population displayed the greatest variation, likely reflecting the influence of multiple invasion sources. These results show that wing morphology can reliably detect invasion-related patterns and phenotypic plasticity and is a valuable tool for biomonitoring in integrated pest management programs, especially in areas where genetic methods are less feasible. This study highlights the utility of geometric morphometrics in monitoring the spread and adaptation of invasive species such as *H. halys* in different environments.

ATE230025 in the form of a grant`s collaborations expenditures for HAB. The specific roles of this author are articulated in the `author contibutions` section.

## Introduction

The brown marmorated stink bug (*Halyomorpha halys* (Stål, 1855)) of the Pentatomidae family is a globally important invasive pest and poses a significant threat to agricultural production [1,2]. The damaging effects of this pest are intensified by its polyphagous nature: it attacks over 300 known host plants and causes considerable damage to field, fruit and vegetable crops as well as ornamental plants [3–6]. Damage is caused by both adults and nymphs through their piercing-sucking feeding mode, during which they inject digestive enzymes into vegetative and generative plant parts, breaking down plant tissues and subsequently extracting the pre-digested plant material. The adults feed mainly on fruits, while the nymphs tend to remain on leaves and stems, resulting in characteristic damage such as deformation and subsequent necrosis of the affected plant tissue [1,7].

*Halyomorpha halys*, originally native to Eurasia, over the past three decades, it has spread rapidly through various pathways, including air transportation, cargo shipments, plant trade, and vehicles [1,8]. First discovered in the mid-1990s, it has since spread widely across North America, causing millions of dollars in damage to fruit and horticultural crops over the past two decades, including $37 million in losses to mid-Atlantic apples during severe pest pressure in 2010 and with some stone fruit growers losing over 90% of their crop [9]. In Europe, *H. halys* was first discovered in Liechtenstein in 2004 [10] and has since expanded its presence across almost all European countries, excluding the Scandinavian and Baltic regions. It was discovered in Greece in 2011 [11], in Serbia in 2015 [12] and in 2017 in both Croatia [13] and Slovenia [14]. In Italy, it emerged a major pest in fruit production in 2019, causing an estimated economic loss of around 600 million euros [15]. Its successful global invasion is driven by its adaptability, polyphagy, high reproduction rate, destructive feeding behavior, and high mobility—both through active flight over long distances and passive human-mediated transport [16,17].

Genetic variability is a fundamental driver of species diversity and adaptation to new habitats, shaping phenotypic traits and influencing invasion dynamics, providing critical insights into the mechanisms of successful invasions [18,19]. In the context of invasive species like *H. halys*, genetic variability plays a role in enabling populations to adapt to diverse environments and expand into new habitats, ultimately driving their invasion success [20]. Genetic analyses of *H. halys* populations, particularly using microsatellite markers and mitochondrial genes (cytochrome c oxidase I (COI) and cytochrome c oxidase II (COII)), have provided valuable insights into the pest's invasion pathways and genetic diversity in different regions [21–24]). Studies have revealed significant differences in haplotype diversity between native populations in Asia and those in newly invaded areas such as North America and Europe [21,23]. The complexity of the global spread of *H. halys* is highlighted by the identification of multiple introductions into Europe and North America, with different genetic profiles indicating separate invasion events [24,25]. The spread of *H. halys* in North America and Europe was found to have occurred predominantly from China, with the exception of Greece, where populations were introduced from Korea [26]. The spread of *H. halys* in Europe is attributed to multiple invasions from different Asian regions and secondary invasions between European countries, possibly favoured by a bridgehead effect. Once established in local areas, *H. halys* spread by both active dispersal and passive transport [27–30].

Despite advances in understanding invasion pathways through genetic markers, these methods alone may not fully capture the extent of morphological variability, which can provide complementary insights into the adaptability of pests. Previous research has shown that morphological differences can provide valuable insights into pest adaptations to agroecological conditions and invasion pathways, as demonstrated in studies on the corn rootworm [31], sugar beet weevil [32] and wireworms [33].

The morphometric markers can serve as biomarkers that may reveal changes in the genetic structure of populations through phenotypic changes [34]. Wings are frequently used to detect population changes due to their inherent stability and transparency [35]. As wing shape and size are highly responsive to agroecological and genetic pressures, they are often among the first physical traits to change, facilitating the rapid dispersal of species [34]. These traits can complement, and in some cases even replace, genetic markers for detecting shifts in population structure and assessing invasion success [31,32,36]. Despite its recent establishment in Europe, the morphological variation of *H. halys* in these regions remains poorly understood, highlighting the need for research to elucidate its invasion patterns.

The aim of this study is to investigate the phenotypic variability of *H. halys* populations in southern Europe using geometric morphometric techniques, focusing on variation in wing shape and size. By assessing the relationship between these morphological traits and the pest's dispersal patterns, we aim to better understand how these traits contribute to its invasion potential. In addition, this study will investigate how geometric morphometric markers, rather than population genetic methods, can be used to describe patterns associated with the biological invasion of *H. halys* to provide a cost-effective and practical tool for monitoring this invasive species.

## Materials and methods

### Ethics statement

*Halyomorpha halys* is an invasive and established pest in Southern Europe. No special permission was needed for its collection in this study.

### Sample sites and specimen collection

The study focused on the assessment of morphological variability of *H. halys* populations from southern Europe and was conducted in four countries: Croatia, Greece, Serbia and Slovenia (Table 1). Adult specimens were collected from agroecological sites in regions where this pest first appeared in masses. In Croatia and Serbia, collections were conducted in 2019, in Slovenia in 2021 and in Greece in 2022. This approach aimed to capture the variability and adaptability of *H. halys* in different environmental and temporal contexts and to gain insights into its morphological plasticity in the region.

The collected specimens (N = 540) were preserved in 70% ethanol, and sex was determined by examination of the abdominal apex before dissection of the wings. Both the left and right anterior and posterior wings were removed from each individual and mounted on microscope slides using the fixative Euparal (Carl Roth GmbH + Co. KG, Karlsruhe, Germany), following standard methods [37]. Detailed information on the sampling sites and the total number of specimens included in the study per country is given in Table 1.

### Data curation and geometric morphometric analyses

A total of 2160 wings of *H. halys*, mounted on microscope slides, were photographed individually with a Nikon D780 camera. The images were saved in JPG format, sorted by location

Table 1. Details of the sample sites and specimens used for morphological analyses.

| Country | Locality | Long | Lat | Males | Females | Population (*n*) |
|---|---|---|---|---|---|---|
| Croatia | Drenčec | 45.83789 | 16.20039 | 70 | 70 | 140 |
| Greece | Episkopi | 40.68992 | 22.12764 | 50 | 50 | 100 |
| Serbia | Bački Petrovac | 45.32101 | 19.70359 | 100 | 100 | 200 |
| Slovenia | Šempeter pri Gorici | 45.93239 | 13.64447 | 50 | 50 | 100 |

and then converted to TPS format using TPSUtil v1.81 software. Subsequently, the software tpsDig2 v2.31 [38] was used to digitize 16 landmarks on the anterior wings and 15 on the posterior wings (Fig. 1), which were defined by vein branches or vein ends (S1 file).

To determine whether size had an impact on the data, multivariate regression analysis was performed to examine the differences between genders, with centroid size as the independent variable and shape as the dependent variable. A violin diagram was calculated using the ggplot2 package in R software for graphical comparison of the variation in centroid size. Finally, to evaluate the shape between possible routes of introduction a mean shape was calculated using the country classifier covariance matrix, and the data were used to superimpose and highlight differences in landmark variation between groups and visualise in an averaged PCA. All morphometric data were first analysed using MorphoJ v1.06d [40] and subsequently statistical analyses such as (PCA, CVA) were performed using the software R [41] with the morphometric package geomorph [42].

## Results

The data were first evaluated by calculating the measurement error by double digitizing the landmarks. A Procrustes ANOVA revealed that the mean squares (MS) of the error were significantly lower than the mean squares of the individual variation, confirming the reliability of the digitizing process. For the anterior wing, principal component analysis (PCA) showed that the first three components explained 55% of the total shape variation (PC1: 31%, PC2: 13%, PC3: 11%). For the posterior wing, 78% of the shape variation was accounted for by the first three components (PC1: 46%, PC2: 24%, PC3: 9%).

As the data were influenced by sexual dimorphism, the data set was first separated by sex to analyze the variance by country. The PCA results showed that the data largely overlapped, but some points showed significant variance within morphology. To further identify shape variation between countries, a covariance matrix was calculated using the averaged data. The anterior wing analysis revealed that Croatian specimens exhibited the greatest variation compared to other populations, particularly at landmarks LM1 and LM2 near the wing base. Croatian wings were wider, with broader anatomical structures at peripheral landmarks, whereas populations from other countries had narrower wings (Fig. 2).

In males, the centroid size distribution was more homogeneous for both anterior and posterior wings. The greatest variation in size was found in specimens from Slovenia and Croatia, with the smallest individuals coming from Slovenia. The wing size range was moderately high (Fig. 8C, 8D).

## Discussion

In this study, we investigated the wing morphology (size and shape) of *H. halys* populations at a broad geographic scale and evaluated whether morphometric methods can serve as an alternative to population genetic methods to describe invasion-related patterns. The results demonstrated that the shape and size of *H. halys* wings effectively reflect patterns associated with the invasion process, revealing distinct morphological differences between populations. These findings align with population genetic studies conducted in the same regions, which also identified variations linked to invasion dynamics [24,25,28].

For this study, *H. halys* specimens were collected from four southern European countries in agricultural areas, covering different stages of population establishment: from the early establishment phase, characterized by localized presence in Croatia, to the complete establishment phase, marked by widespread distribution and consistent population dynamics in Slovenia, Serbia, and Greece 43-47]. With 2,160 wings analyzed, this study provides one of the

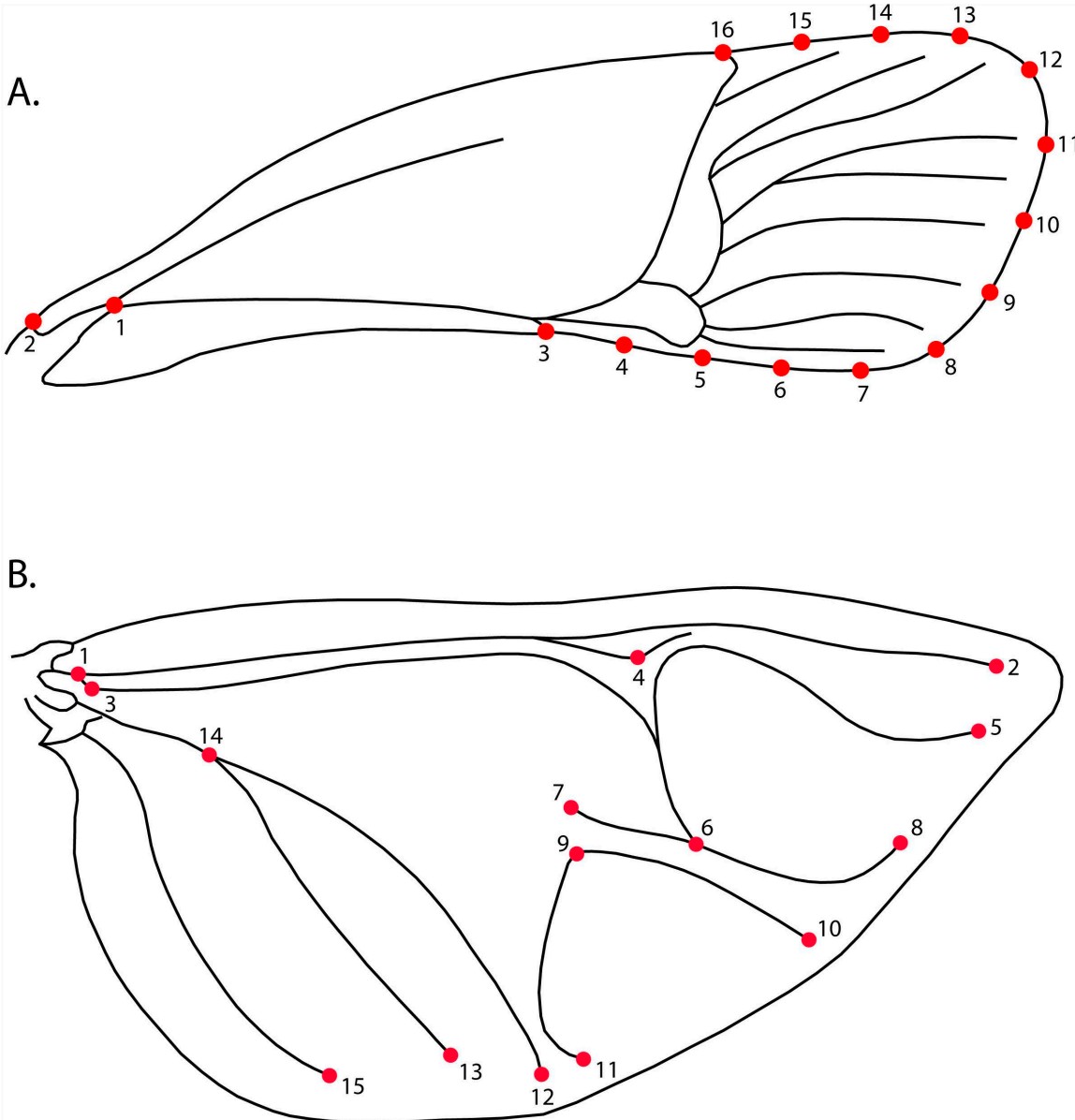

**Fig. 1. Graphical representation of the wing shape of *Halyomorpha halys*.** A: Anterior wing landmarks (16), B: Posterior wing landmarks (15). Landmark coordinate data for anterior and posterior wings were processed using Generalized Procrustes Analysis (GPA) [39], which standardized the digitized individuals by removing variation in size, position, and orientation, aligning them based on centroid size. Variation in wing shape was then analysed by projecting the data using Principal Components Analysis (PCA) based on the covariance matrix of each specimen. Data were then split by sex to calculate differences for each sex and wing. To visually examine shape variation among countries, a Canonical Variate Analysis (CVA), a type of discriminant analysis designed for more than two groups, was conducted. Mahalanobis distances were calculated to compare morphological differences among the four countries and were statistically tested for significance using a permutation test with 1,000 iterations. Mahalanobis distances are a multivariate statistical measure used to assess the difference between groups by considering the covariance structure of the data. In geometric morphometrics, they quantify shape differences between groups while accounting for the correlation among shape variables, providing a more accurate discrimination of group-specific traits.

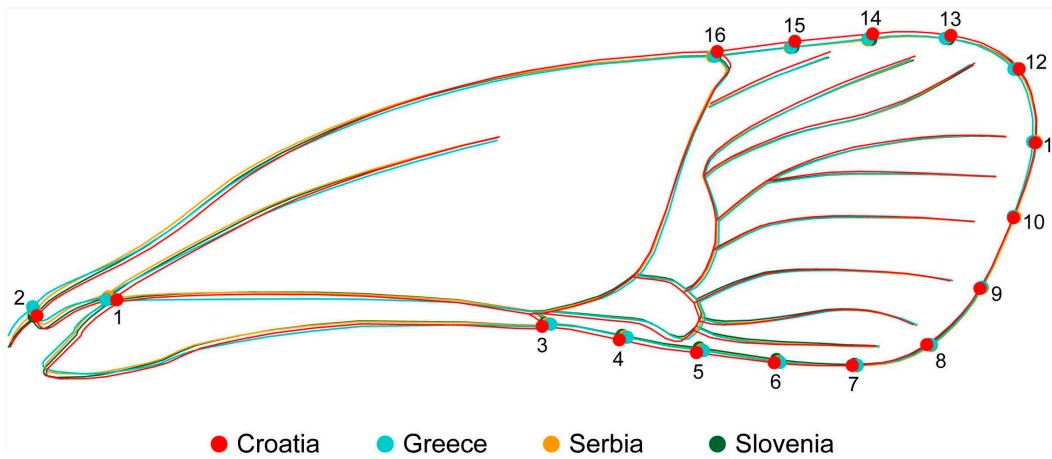

**Fig. 2. Average shape by country classifier of the anterior wing in *Halyomorpha halys*.** Greater variation in the anatomical shape of the posterior wings was observed, with different patterns emerging in the countries studied. The Croatian specimens had the widest wings, with a clear differentiation of the outline and the LM1, LM2 and LM3 landmarks, which distinguishes them from the wings of other countries. In contrast, the wings from Greece showed more variation in the central veining, especially around LM9 and in the lower posterior region at LM15. The wings from the other countries showed more similarity to each other and were narrower than those from Croatia and Greece (Fig. 3).

most extensive morphological datasets for *H. halys*, capturing detailed phenotypic variations along its geographical invasion corridor through southern European agricultural regions. Furthermore, the data shown that population differences identified by mitochondrial genes [24, 25] can also be detected at the same geographical locations by morphometric methods.

The results presented here clearly distinguish populations from Slovenia, Croatia and Serbia from those in Greece, which was one of the first regions in southern Europe to be invaded by *H. halys* [11,44–47]. The differences in wing shape and size observed in this study, which align with a south-to-north and west-to-east dispersal pattern (Fig. 7), are consistent with the genetic findings of previous studies [4,24]. These genetic studies revealed that the Greek population originated from multiple sources, including direct introductions from China, Japan, and Korea, as well as established populations in the USA [22,48,49]. The Slovenian population is likely derived from neighboring Italy, where a widespread and highly invasive population was documented in 2015, causing significant agricultural damage [4,50].

Serbia has the highest morphological interpopulation variation among the populations analyzed in this study (Figs 4 and 5). The genetic diversity values reported for Serbia [24] were lower than those observed in Greece [22,28]. Our results confirm that the *H. halys* populations in Serbia, which have comparable diversity levels to regions with multiple invasion sources such as Greece, likely originate from multiple source populations in surrounding European countries [44].

Genetic studies [24,25] have reported very low genetic diversity in Slovenia and especially Croatia, primarily based on mitochondrial markers COI and COII. This finding contrasts with the results of our study, which observed significant morphological variation in these populations. A possible explanation for this discrepancy could be the small sample sizes used in the genetic studies conducted so far. Previous study [51] analyzed specimens of *H. halys* from a broad geographic range in China, Korea, and Japan, but the limited sample size remains a challenge. They found that genetic diversity in the U.S. populations was extremely low, with only two mitochondrial haplotypes identified, and half of the populations exhibited zero genetic diversity, with only a single haplotype detected. The authors concluded the

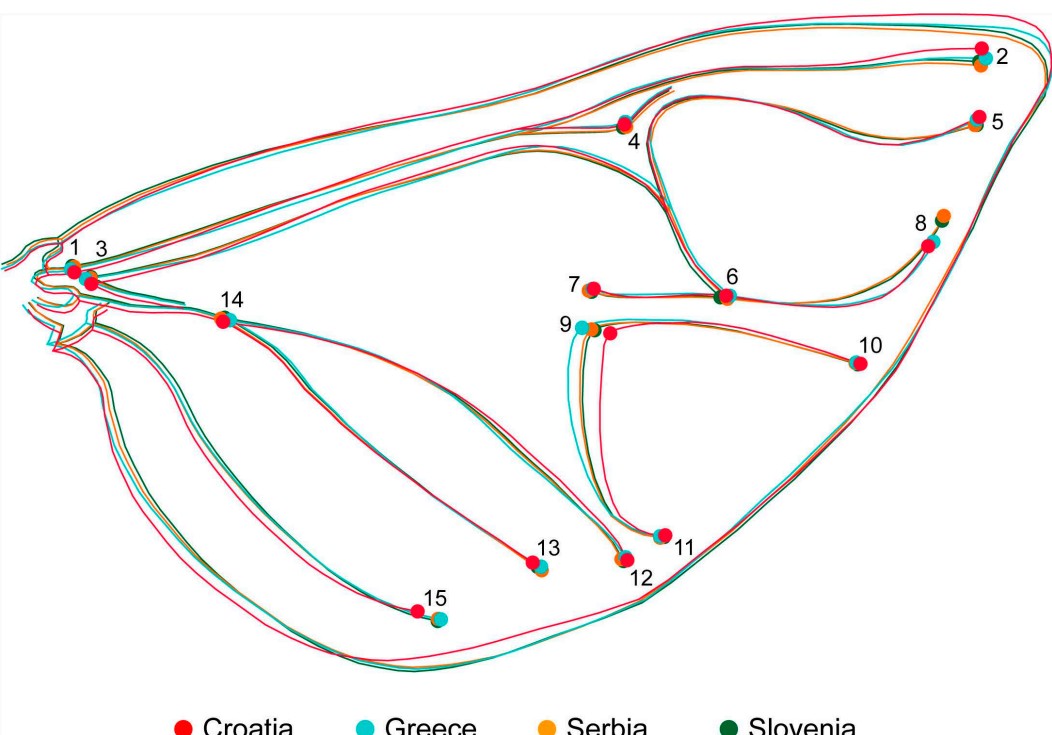

**Fig. 3. Average shape by country classifier of the posterior wing in *Halyomorpha halys*.** To better represent and highlight the differences between the populations, a Canonical Variate Analysis (CVA) was used with "Country" as the classifier, focusing on maximizing the variance between the groups. The CVA showed that the Croatian population (represented by red dots) differed the most from the other countries. In contrast, the Slovenian population showed less dispersion between individuals, while the Serbian population showed overlap with both the Greek and Slovenian populations but showed a higher degree of variation within the population. These results reflect morphological diversity between populations (Fig. 4).

importance of using adequate sample sizes to accurately capture genetic diversity and population dynamics, particularly when studying invasive species. Previous study [52] demonstrated that under-sampling can lead to inaccurate estimates of genetic diversity, and they recommend sampling a larger number of individuals (n ≥ 25 as a general rule) from a population to document the presence of lower frequency mitochondrial haplotypes, which was not the case in genetic studies of *H. halys* from Slovenia and Croatia. The relationship between mitochondrial haplotypes and phenotypic variability in wing morphology is influenced by indirect mechanisms. Mitochondrial genes, such as COI and COII, primarily regulate metabolic and energy production processes, which can indirectly affect developmental traits like wing shape and size by influencing energy allocation during growth [53,54]. Additionally, mitochondrial haplotypes may reflect historical adaptations to local environmental conditions, aligning with regional morphological traits optimized for ecological pressures. Nuclear-mitochondrial interactions could further contribute to phenotypic variability, as disrupted compatibility between the genomes might affect developmental pathways [55].

The period between the first documented introduction and sampling provides insights into the invasion patterns of *H. halys* populations in southern Europe. In Greece, *H. halys* was first introduced in 2011 and economic damage occurred seven years later, in 2017 [43]. In Serbia, the period between the introduction and the occurrence of economic damage was shorter and lasted two years from 2015 to 2017, a similar pattern was also observed in Slovenia (2017–2018). In contrast, *H. halys* was introduced to Croatia in 2017, with the first mass population

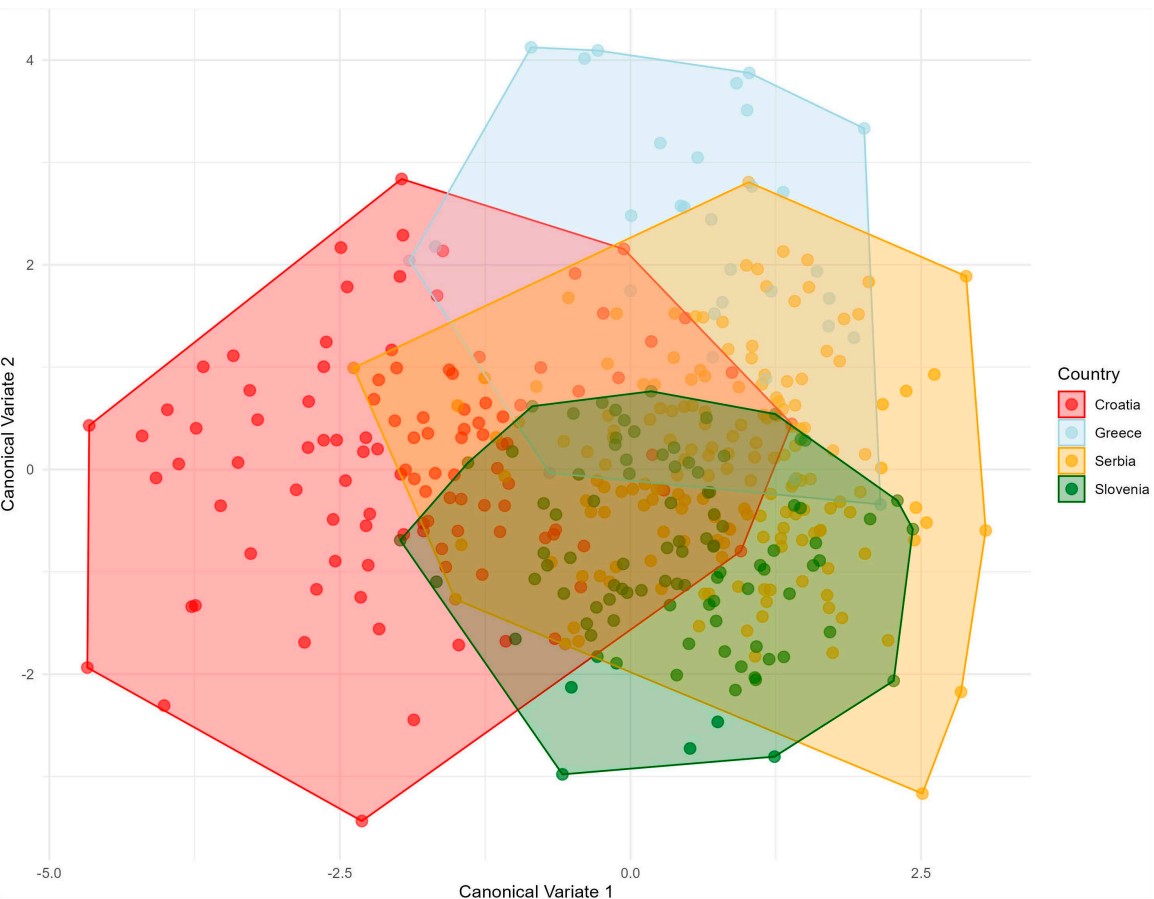

**Fig. 4. Canonical variate analyses of the anterior wing of *Halyomorpha halys* between countries, colours represent the different countries: red: Croatia, light blue: Greece, yellow: Serbia and green: Slovenia.** The Canonical Variate Analysis (CVA) for the posterior wing showed a separation of the Greek specimens, which were predominantly located in the upper left quadrant of the scatter diagram, indicating different morphological features. In contrast, there was overlap in shape variation between the Slovenian and Serbian populations, with some divergence occurring in the Serbian specimens on the left side of the diagram. As in the analysis of the anterior wings, the Croatian specimens showed the most pronounced morphological differentiation from the other three countries, emphasising their pronounced variation in wing shape (Fig. 5).

sampled in 2019 [46]. However, economic damage was not observed until 2024, indicating a seven-year delay between introduction and significant impact [56]. This longer time span in Croatia could indicate a slower adaptation to local environmental conditions or that the population was still being in the establishment phase at the time of sampling. The shorter periods observed in Serbia and Slovenia could indicate a faster adaptation to the local environmental conditions. In Croatia, however, the population was sampled during its first mass emergence before it caused economic damage, possibly indicating ongoing adaptation.

The Mahalanobis distances presented here (Table 2) reveal significant morphological divergence among the studied populations and provide insights into the origin and dispersal dynamics of *H. halys*. The smallest distances are observed between the populations of Serbia and Slovenia, particularly for the anterior wing (1.2186) and posterior wing (1.7352), indicating high morphological similarity. Distances around 1.5 suggest strong morphological similarity, likely reflecting recent gene flow or shared ancestry. By contrast, Croatia exhibits greater divergence, especially from Greece, with the largest distances recorded between Croatia and Greece for both the anterior wing (2.7112) and posterior wing (3.9881). Distances

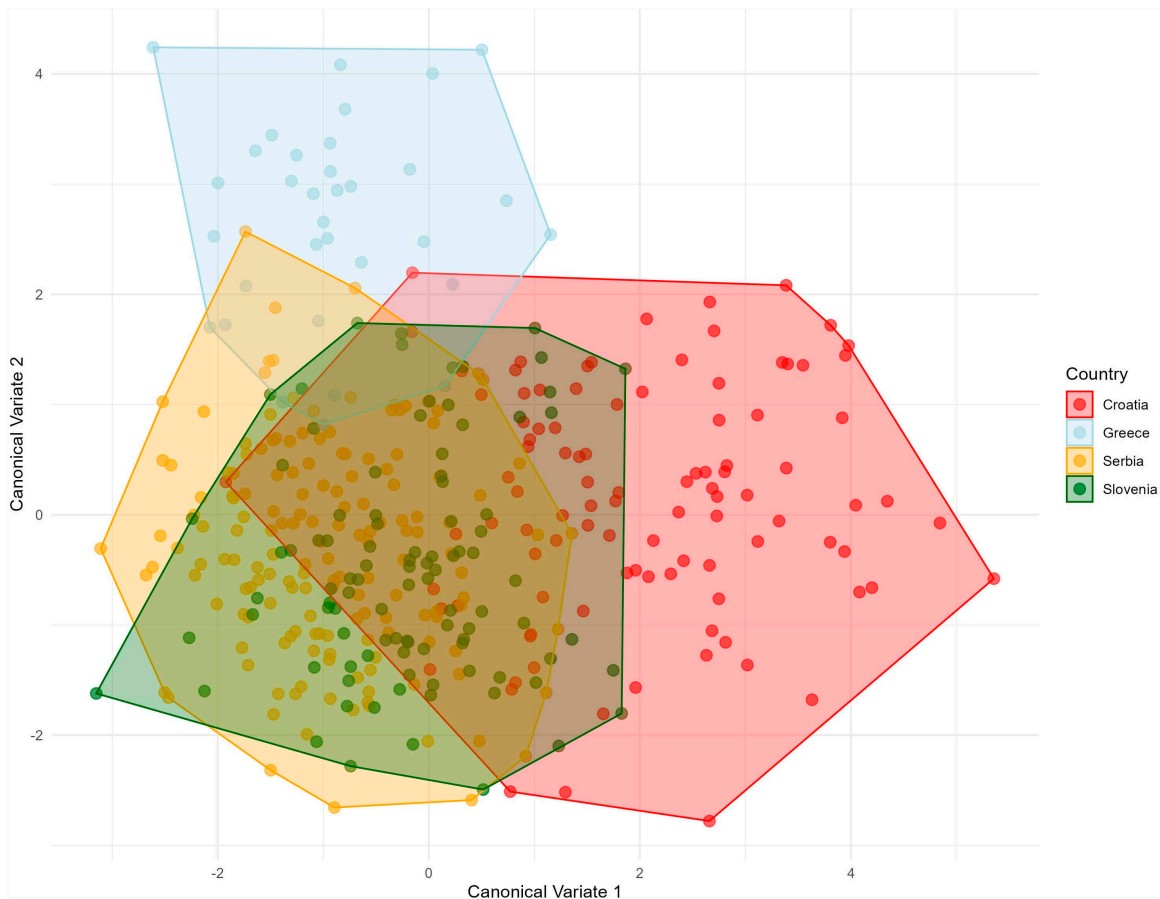

**Fig. 5. Canonical variate analyses of the anterior wing of *Halyomorpha halys* between countries, colors represent the different countries: red: Croatia, light blue: Greece, yellow: Serbia and green: Slovenia.** To evaluate the allometric influence on wing shape, a multivariate regression analysis was performed for both sexes and wing types. A small but statistically significant allometric effect was found for the anterior wings of the female specimens, explaining 4.72% of the shape variation (p < 0.001). In the posterior wings of females, a higher percentage of shape variation was associated with allometry, accounting for 10.8% (p < 0.001). Although the allometric effects were statistically significant, they did not substantially influence the overall shape variation observed, suggesting that other factors, such as genetics, may have a more prominent role in driving morphological differences (Fig. 6).

exceeding 2.5 reflect substantial morphological differences, which may result from distinct environmental pressures, genetic isolation, or independent invasion events. Similarly, Greece shows divergence from Serbia and Slovenia, with distances ranging from 2.5 to 3.9, supporting the hypothesis of a unique invasion pathway or prolonged isolation. These findings suggest that Serbia and Slovenia may represent populations closer to the primary invasion source, while Greece and Croatia exhibit greater morphological variability, possibly due to secondary invasion events or adaptive responses to local ecological conditions.

The invasion pattern presented can be explained by phenotypic plasticity, defined as the ability of a single genotype to produce different phenotypes in response to different environmental conditions [57], which influences the adaptability of invasive species. The population from Greece, which was the first to be introduced to southern Europe, shows signs of lower phenotypic plasticity, as evidenced by its smaller wing size and more uniform shape. This may indicate that the population has adapted to local environmental conditions and no longer relies on a broad range of phenotypic responses. The multivariate regression analysis also revealed a consistent allometric influence on both wing types, with the Greek

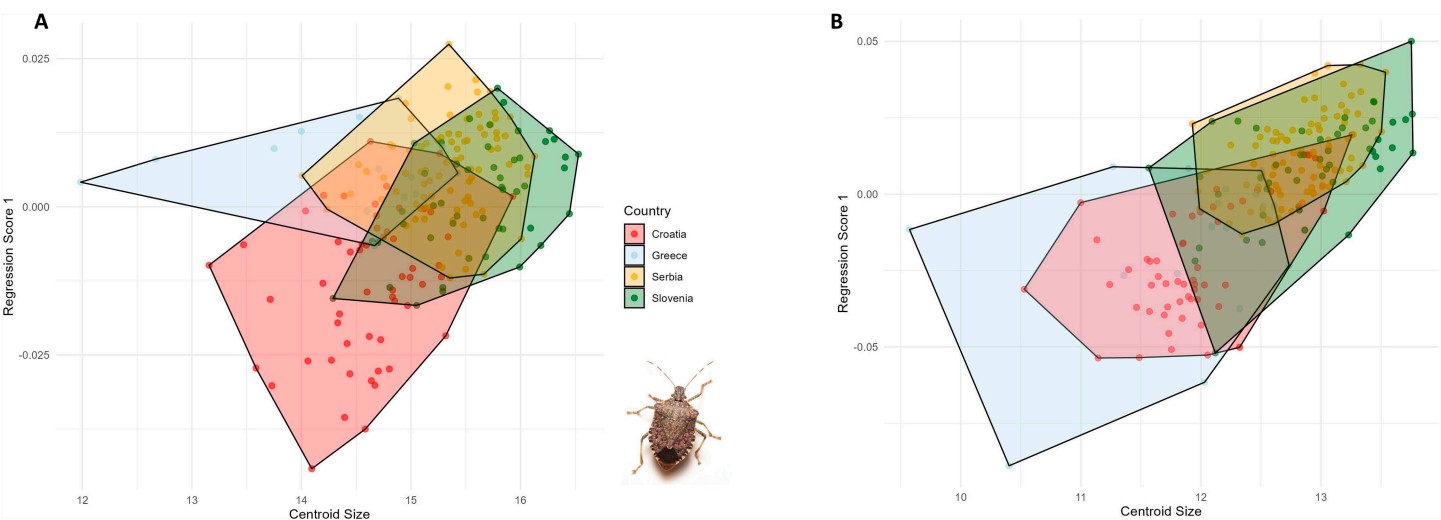

**Fig. 6. Multivariate regression of the female wing shape (dependant) vs centroid size (independent) of *Halyomorpha halys*.** A: Anterior wing, B: Posterior wing. In male wings, allometry accounted for 4.5% of the shape variation in the anterior wings (p < 0.001). In the posterior wings, allometric effects were slightly more pronounced than in females, explaining 13.8% of the shape variation (p < 0.001). While statistically significant, these effects were modest and insufficient to significantly influence the overall shape variation patterns observed in the population (Fig. 7).

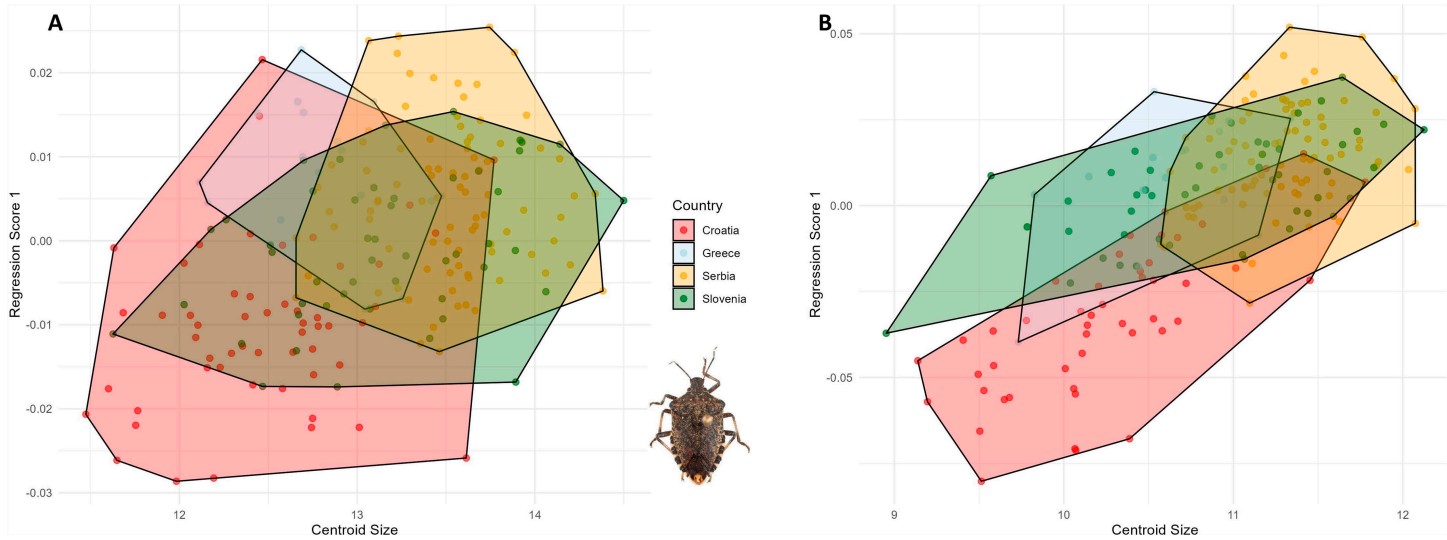

**Fig. 7. Multivariate regression of the male wing shape (dependant) vs centroid size (independent) of *Halyomorpha halys*.** A: Anterior wing, B: Posterior wing. The multivariate regression analysis showed a consistent allometric effect across both wing types, with a lower percentage of the allometric effect observed in the anterior wings and a slightly higher percentage in the posterior wings. To further investigate the size distribution, the centroid sizes (CS) were extracted for each specimen. In females, the size distribution of the anterior wing centroids showed consistent patterns among populations from Croatia, Slovenia, and Serbia, while the Greek population exhibited greater variation, including smaller specimens. A similar trend was observed for the posterior wing, with Croatian specimens tending to be larger (Fig 8A, 8B).

population showing greater variation in the centroid size distribution, particularly in females, where smaller individuals were more common compared to other populations. Although the centroid size distribution in Greece showed a moderately wide range of wing size in males, it remained more homogeneous compared to the populations in Croatia and Slovenia. These results highlight the unique position of Greece as an early invasion site, where the population

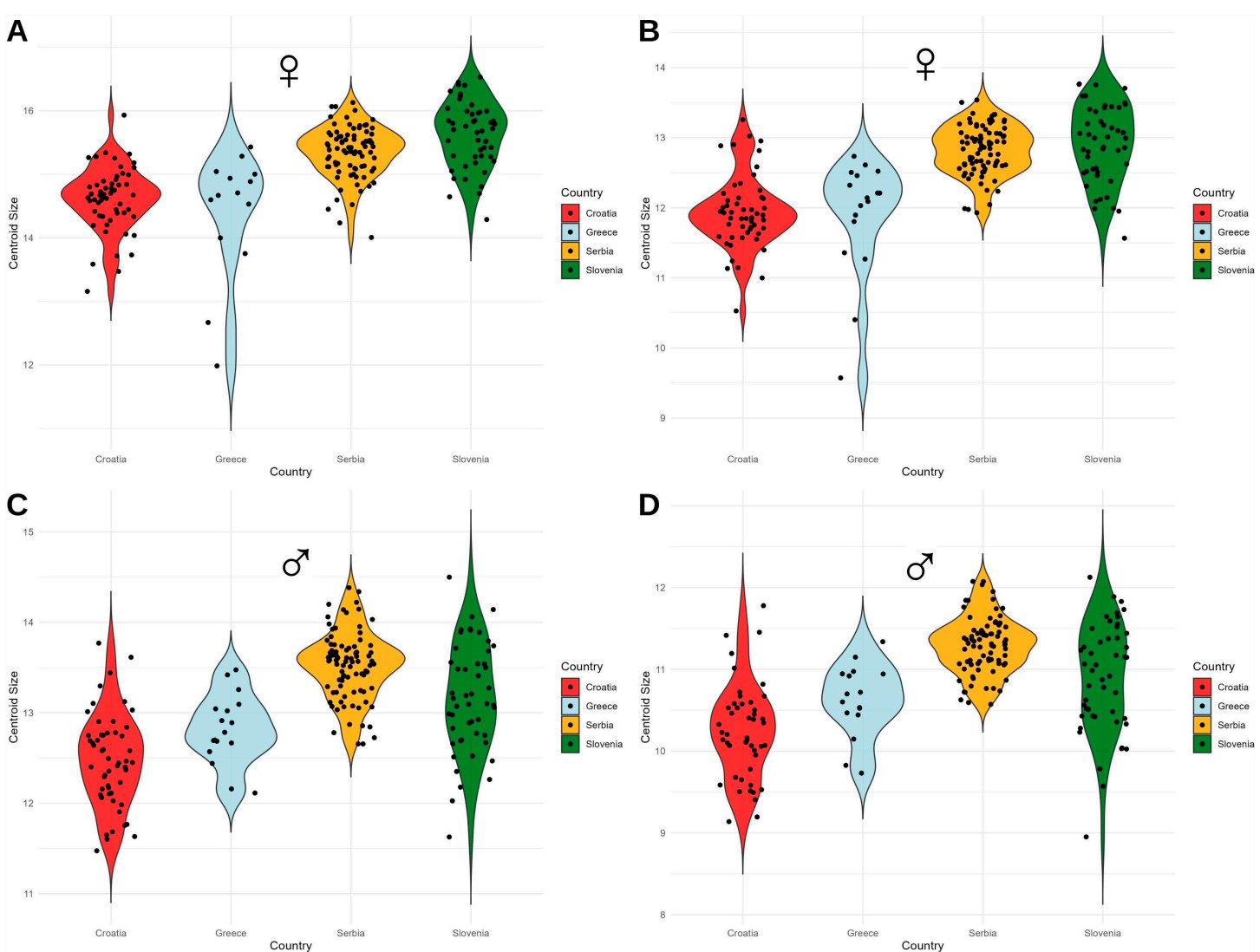

**Fig. 8. Violin graph of the distribution of centroid size between countries in *Halyomorpha halys*.** A: Female anterior wing B: Female posterior wing C: Male anterior wing D: Male posterior wing. The average PCA (Fig. 9) graphically represents the Mahalanobis distances and their associated permutations (corrected for multiple comparisons), highlighting significant relationships between the groups (Table 2). Pairwise comparisons revealed the greatest Mahalanobis distances between Greece and all other countries, reflecting pronounced differences in anterior and posterior wing shape (Table 2). The smallest distances were observed between Serbia and Slovenia, indicating closer morphological similarity between these populations. The Croatian population displayed mixed variation, sharing the first principal component in the morphospace with both Serbia and Slovenia (Fig. 9).

exhibits a narrower centroid size distribution, potentially indicating reduced phenotypic plasticity and a higher degree of specialization to its local environment.

In contrast, the Serbian population shows the highest variation between populations, with greater variability in wing shape and size. This suggests that the Serbian population may have been influenced by multiple sources of invasion, resulting in greater genetic and phenotypic diversity. Such diversity likely enhances the population's ability to adapt to a wider range of environmental conditions, suggesting higher phenotypic plasticity.

The Slovenian population shows moderate phenotypic plasticity, with less variation than the Serbian and Croatian populations, suggesting ongoing adaptation, albeit to a lesser extent than in Serbia. The Croatian population, with its variability in wing morphology,

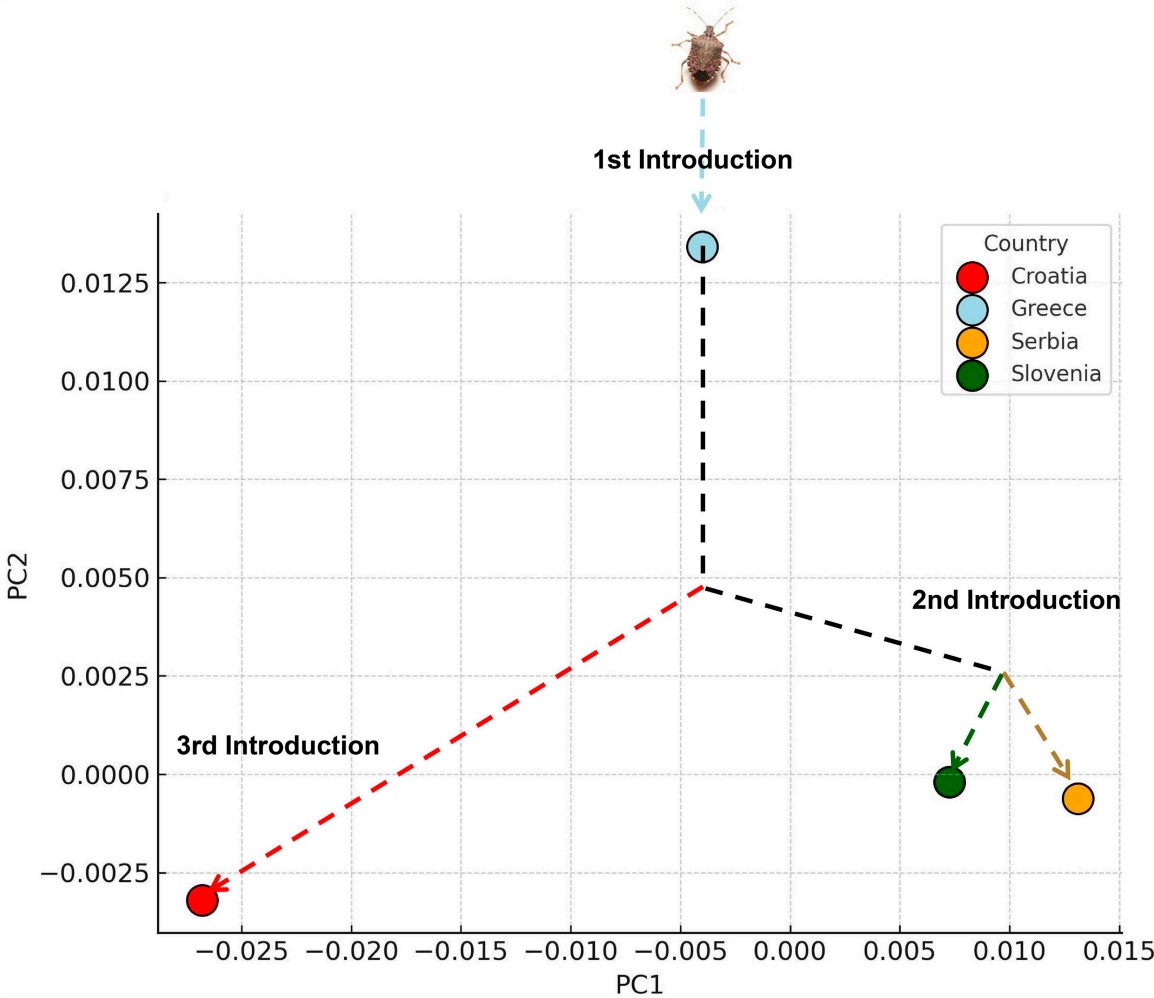

**Fig. 9. Principal Component of the average body shape between countries, the average represents the possible ways of introduction of** *Halyomorpha hayls*. Arrows represent the pathway of shape possibilities in an average shape space.

may have higher phenotypic plasticity, suggesting that it is still adapting to local conditions or is subject to more diverse ecological pressures that maintain this variation. This higher plasticity could explain the delayed onset of economic damage compared to neighbouring regions.

The results of this study provide further evidence that phenotypic differences tend to evolve faster than genotypic [34]. Population genetic studies of *H. halys* in Europe [22,24,25,28] indicate that haplotypes in newly invaded areas have not yet stabilized sufficiently within populations to serve as consistent biomarkers for surveillance. Several factors, such as genetic drift or natural selection, may explain this. However, phenotypic variation has been commonly observed in populations where genetic markers did not reveal differences [58–60]. Although both genetic and phenotypic markers have different applications, phenotypic differences may prove more suitable as population markers for biomonitoring [18,19,31,34,61], especially when the aim is to monitor invasive populations and invasive patterns efficiently. Phenotypic markers, such as wing shape, are valuable tools for detecting and interpreting environmental changes because they often reflect both genetic and environmental influences on an organism [32,36,62–65]. In the case of *H. halys*, variations in wing

**Table 2. Mahalanobis distances between countries with respective permutation comparison p-values.**

| Mahalanobis Distances p-value | Croatia | Greece | Serbia |
|---|---|---|---|
| **Anterior Wing** | | | |
| Greece | 2.7112 <0.0001 | | |
| Serbia | 2.63 <0.0001 | 1.866 <0.0001 | |
| Slovenia | 2.3961 <0.0001 | 2.4862 <0.0001 | 1.2186 <0.0001 |
| **Posterior Wing** | | | |
| Greece | 3.9881 <0.0001 | | |
| Serbia | 2.63 <0.0001 | 3.0095 <0.0001 | |
| Slovenia | 2.4753 <0.0001 | 3.2761 <0.0001 | 1.7352 <0.0001 |

morphology may provide specific insights into how populations respond to ecological pressures, such as climate, habitat structure, and resource availability.

Testing phenotypic differences on a larger geographical scale, especially including all stages of the *H. halys* invasion process, would further validate this approach. The results of this study emphasise the importance of phenotypic markers for invasive species management, especially when genetic techniques are too expensive. This research has shown that differences in shape and size can reliably reveal population variation and could serve as an effective, cost-efficient biomonitoring tool.

## Supporting information

**S1 File. S1 Halyomorpha halys data set.**
(TXT)

## Author contributions

**Conceptualization:** Martina Pajač Beus, Darija Lemic, Ivana Pajač Živković.

**Data curation:** Martina Pajač Beus, Hugo A. Benitez, Mojca Rot, Aleksandra Konjević, Stefanos S. Andreadis.

**Formal analysis:** Hugo A. Benitez, Laura M. Perez, Aleksandra Konjević.

**Investigation:** Martina Pajač Beus, Mojca Rot, Stefanos S. Andreadis, Ivana Pajač Živković.

**Methodology:** Martina Pajač Beus, Darija Lemic, Mojca Rot, Aleksandra Konjević, Stefanos S. Andreadis, Ivana Pajač Živković.

**Project administration:** Ivana Pajač Živković.

**Software:** Hugo A. Benitez, Laura M. Perez.

**Supervision:** Darija Lemic.

**Validation:** Hugo A. Benitez, Mojca Rot, Aleksandra Konjević, Stefanos S. Andreadis, Ivana Pajač Živković.

**Visualization:** Hugo A. Benitez, Laura M. Perez, Ivana Pajač Živković.

**Writing – original draft:** Martina Pajač Beus, Darija Lemic, Hugo A. Benitez, Laura M. Perez, Ivana Pajač Živković.

**Writing – review & editing:** Martina Pajač Beus, Darija Lemic, Hugo A. Benitez, Mojca Rot, Aleksandra Konjević, Stefanos S. Andreadis, Ivana Pajač Živković.

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
