## [Decision Letter · Decision Letter 0]

15 Dec 2024

PONE-D-24-53394Tracing the Invasion: Wing Morphometrics Reveal Population Spread and Adaptation Patterns of Halyomorpha halys (Stål, 1855) Across Southern EuropePLOS ONE

Dear Dr. Lemic,

Thank you for submitting your manuscript to PLOS ONE. After careful consideration, we feel that it has merit but does not fully meet PLOS ONE’s publication criteria as it currently stands. Therefore, we invite you to submit a revised version of the manuscript that addresses the points raised during the review process.

We look forward to receiving your revised manuscript.

Kind regards,

Vazrick Nazari, PhD

Academic Editor

PLOS ONE

Journal Requirements:

4. In the online submission form, you indicated that The data underlying the results presented in the study are available from corresponding authors. 

Reviewers' comments:

Reviewer's Responses to Questions

**Comments to the Author**

1. Is the manuscript technically sound, and do the data support the conclusions?

Reviewer #1: Yes

Reviewer #2: Yes

2. Has the statistical analysis been performed appropriately and rigorously? 

Reviewer #1: Yes

Reviewer #2: Yes

3. Have the authors made all data underlying the findings in their manuscript fully available?

Reviewer #1: Yes

Reviewer #2: Yes

4. Is the manuscript presented in an intelligible fashion and written in standard English?

Reviewer #1: Yes

Reviewer #2: Yes

5. Review Comments to the Author

Reviewer #1: Overall appreciation of the manuscript:

This journal article draft presents an interesting and relevant investigation into the morphological wing variations of Halyomorpha halys populations across southern Europe. While the study successfully explores the potential of wing geometric morphometrics as a cost-effective alternative to genetic methods, several areas could benefit from improvement.

The manuscript is somewhat unconcise in its results section, where results are interpreted, discussed and linked to speculations. Additionally, the manuscript the discussion is rather repetitive, particularly regarding the role of phenotypic markers, which can be streamlined to improve clarity.

Speculative statements, such as suggesting lower phenotypic plasticity in the Greek population without direct supporting evidence, weaken the overall argument and should be more cautiously framed. Additionally, the manuscript could do more to connect the morphological findings with genetic data, offering a clearer explanation of how these two aspects interact (and not interact). Some sections also require a more detailed explanation of the methodology and a stronger justification for the conclusions drawn, especially when discussing the potential role of genetic drift or adaptation. Overall, while the study contributes valuable insights, the draft would benefit from tighter organization, clearer and concise explanations, and more careful interpretation of the results.

Comments, questions and suggestions to be addressed:

Line 28-29: (suggestion for readability): analysing the size and shape morphology of the anterior and posterior wings

Line 32: Specify exact number of specimens and wings analysed

Line 52: “Harmful effects” are not ascribed to a pest but rather to a chemical, please rephrase this sentence

Line 55: Technically stink bugs like H. halys have a piercing-sucking mode of feeding, not plant juices are sucked, but more specifically pre-digested plant mass after injecting digestive enzymes into plant tissues. Please specify this better

Line 61: (suggestion for readability): Over the past three decades, it has spread rapidly through various pathways, including air transportation, cargo shipments, plant trade, and vehicles.

Line 63: “significant spread” what is meant by this?

Line 63: specify “the millions of dollars of damage” with a cited example, otherwise this is a vague statement

Line 66: Might be slightly over stated, not in Scandinavian and Baltic countries.

Line 68: (suggestion for readability): became => emerged

Line 70: Successful global invasion not only due to its ability to adapt but also (and mainly) its polyphagy, high mobility (active and passive)

Line 71-73: threat and invasion rate causes are mixed here rephrase with Line 70 to clarify why it has had invasive success and poses a major threat.

Line 74: Vague statement and a missed opportunity to connect genetic variability directly to the phenotypic observations or invasion dynamics

Suggestion:

In the context of invasive species like Halyomorpha halys, genetic variability plays a crucial role in enabling populations to adapt to diverse environments and expand into new habitats, ultimately driving their invasion success (Gloss et al., 2016)

Line 75: These studies using COI and COII sequences should be cited at the end of this sentence.

Line 88: Are there more European studies on this?

Line 90: Statement might be overly strong, genetic diversity studies & morphological variability studies would be complimentary

Line 91-94: “numerous studies” and “many others can come across as vague. Replace with more precise statements that conveys specificity and credibility

Suggestion:

Previous research has shown that morphological differences can provide valuable insights into pest adaptations to agroecological conditions and invasion pathways, as demonstrated in studies on the corn rootworm (Mikac et al., 2016) and sugar beet weevil (Lemic et al., 2016).

Line 94-96: “may reveal”

Line 101: Here the statement of complimenting genetic marker studies is made, I refer back to the comment in Line 90. Rephrase this in one of the statements and avoid repetition.

Line 103-105: Too many unnecessary words like ‘indicating a research gap that needs to be filled”

Suggestion:

Despite its recent establishment in Europe, the morphological variation of H. halys in these regions remains poorly understood, highlighting the need for research to elucidate its invasion patterns.

Line 105-107: Also lacks conciseness, clarity and academic tone

Line 108-118: The hypothesis and objectives are restated in slightly different words, making the section unnecessarily repetitive.

Line 127-129: repeating "multi-country, multi-year" can detract from an academic tone

Line 131: When discussing the totality of samples use N = 540 instead of n = 540

Line 142: Sentence too long for readability it should start a new sentence after “camera”. Start new sentence with: The images were saved in JPG format…

Line 148: Unnecessary use of semicolon: connect sentences with a word or split sentence.

Line 152-154: Double use of angle brackets can be avoided by phrasing it like this for instance:

To visually examine shape variation among countries, a Canonical Variate Analysis (CVA), a type of discriminant analysis designed for more than two groups, was conducted.

Line 154-157: The phrase “using permutation test” is missing the article “a” before “permutation test,”. It should be corrected to “using a permutation test”. An additional suggestion to clarify the text could be:

Mahalanobis distances were calculated to compare morphological differences among the four countries and were statistically tested for significance using a permutation test with 1,000 iterations.

Line 159: A comma is needed after “Additionally”

Line 165: then => subsequently

Line 166-167: Correct in-text citation for R is (R Core Team, 2024). Additionally, provide a more concise phrase to mention that the statistical analyses were done in R.

Suggestion:

All statistical analyses were done in R v.4.1.1 (R Core Team, 2024) using packages x and y

Line 174 -175: The explanatory power of the principal components is not consistently rounded (78.00% and 46.00% would then be correct). In this case one decimal place seems enough.

Line 178-183: This section reads more like an interpretation of the results and could better fit in the discussion. Language like “highest variability” and “tended to be wider” is interpretation of the results and needs to be moved to the discussion or results need to be more concisely given such as the covariance matrix and the quantitative variability metrics in figure 3.

Line 185-191: The results lack quantitative data (e.g. Mean wing width, variability metrics) and are given in an overly descriptive way rather than presenting the data concisely and leaving interpretations for the discussion.

Line 193-202:

1. The sentence "The CVA clearly showed that the Croatian population (represented by red dots) differed the most from the other populations" repeats information already implied in the subsequent text about Croatia being distinct.

2. Mentioning PCA in a CVA-focused section creates confusion

3. The narrative includes subjective terms like "clearly showed" and "stood out," which should be avoided in the results section to maintain objectivity.

4. The phrase "showed less variation between groups" is vague without quantitative or statistical evidence.

5. Statements such as "These results reflect considerable morphological diversity" and "Croatia standing out as the most distinct group" move toward interpretation, which is more appropriate for the discussion section.

Line 204-211:

While mentioned in Table 2, this section does not provide numerical values (e.g., Mahalanobis distances, percentage variance explained by the CVs) to substantiate claims of differentiation or overlap.

Line 213-220

1. "small but statistically significant" could be more precise. The magnitude of 4.72% and 10.8% already conveys the effect size.

2. The sentence "Despite these statistically significant results, the allometric effects were not sufficient to significantly alter the overall shape variation observed" is repetitive, as the lack of a dominant allometric influence is already implied.

3. While p-values are reported, the biological relevance of 4.72% and 10.8% of variation explained could be contextualized further.

Line 222-228:

The statement "the primary variation is driven by factors other than size alone" is speculative without direct evidence from the study.

The sentence "this effect was not significant enough to alter the overall pattern of shape variation observed in the population" repeats information already implied by the percentages and p-values.

Suggestion for concise results reporting (applicable for previous and next result sections as well):

In male wings, allometry accounted for 4.5% of the shape variation in the anterior wings (p < 0.001). In the posterior wings, allometric effects were slightly more pronounced than in females, explaining 13.8% of the shape variation (p < 0.001). While statistically significant, these effects were modest and insufficient to significantly influence the overall shape variation patterns observed in the population (Figure 8).

Line 230-237:

1. The description of centroid size distribution is qualitative (e.g., "relatively conservative", "greater variation") without providing actual metrics or statistical support.

2. The term "relatively conservative" is vague and could be replaced with a more specific description of the size distribution.

Line 238-241:

1. "the wing size range of wing size" is repetitive and unnecessary.

2. Terms like "more homogeneous" and "moderately high" are unclear without specific quantitative metrics or context.

Line 243-253:

1. Phrases like "indicating that Greece may be more genetically isolated" and "possibly indicating that they originated from a similar source" are speculative and not directly supported by the study's data.

2. The phrase "a mix variation from the last two" is vague and unclear

Line 262: "reliably capture the patterns associated with the invasion process" is broad and could be more specific, highlighting exactly what patterns were observed (e.g., differentiation, variability).

Line 264: The parenthetical "see: Gariepy et al., 2021; Yan et al., 2021, Cesari et al., 2018" is conversational and less formal. Standard citation style should be used.

Line 266: "early to complete establishment phase" is unclear and could benefit from more precise language.

Line 268: "Over 2,000 wings analysed" could mention the exact number for precision.

Line 268: "Represents one of the most comprehensive" is generic and lacks detail.

Line 271: The mitochondrial genes COI and COII should have been mentioned earlier in the introduction, and would then not necessarily have to be explained in the discussion.

Line 277: Citation style is informal ("see Musolin et al.").

Line 278-279: suggested revision:

These findings align with genetic marker studies (Gariepy et al., 2021; Yan et al., 2021), which demonstrated comparable population differentiation.

Line 281: direction => pattern

Line 282: parenthetical reference to Fig. 7 should be (Fig. 7)

Line 282: Start a new sentence and clarify that this was from genetic evidence: “Greek population was shown to originate from multiple sources”.

Line 285: "Most likely originates" is overly speculative and could be made more precise with supporting evidence. “likely originated from neighbouring Italy”

Line 288: Specify that the interpopulation variation is in the morphological diversity

Line 293: (as discussed in Musoulin et al., 2018) does not need parentheses and the citation contains a typo and is likely: Musolin et. al., 2018

Line 296: The explanation that "small sample sizes" account for the discrepancy is plausible but insufficient. Small sample sizes might affect genetic diversity estimates, but their connection to higher observed morphological variation is not explicitly explained.

There is no discussion of how genetic diversity (or lack thereof) in mitochondrial markers COI and COII is mechanistically linked to phenotypic diversity in wing morphology. Mitochondrial genes primarily affect metabolic functions, so their direct influence on wing shape variability is unclear.

Line 297-299: The statement "too small sample size remains a problem" is overly general. It does not specify what aspect of Xu et al.'s findings is affected by small sample size (e.g., underestimated genetic diversity, limited geographic representation, or insufficient statistical power).

Line 304-317:

The biological significance of these distance values (e.g., thresholds for divergence or similarity) is not explained. Without context, it is unclear what constitutes "small" or "large" distances.

Line 344: The term "less phenotypic plasticity" is used but not quantified or clarified, link to the specific trait is missing (likely the centroid size distribution).

Line 357-373:

The use of long, complex sentences detract from clarity and could be streamlined. Break up longer sentences to improve clarity and readability.

Elaborate on how phenotypic markers, like wing shape, can indicate environmental changes. Provide more specific examples or evidence if available.

Figure 10:

Add variance percentages to the axis labels (e.g., "PC1 (X% variance)" and "PC2 (Y% variance)").

While the arrows indicate pathways of introduction, the reasoning behind these pathways is not explicitly linked to the plot. It's unclear whether these are based on Mahalanobis distances, historical data, or another source.

Reviewer #2: Dear authors,

Congratulations on producing a high-quality manuscript.

Overall, this is a well-prepared and technically sound study that provides valuable insights into the invasion patterns of Halyomorpha halys populations using geometric morphometric analyses. The research is comprehensive, with robust sample sizes and appropriate methodologies, making a meaningful contribution to the field of invasive species monitoring and management.

Concerns about dual publication, research ethics, or publication ethics in this manuscript were not identified. The study appears original, adheres to ethical guidelines, and is appropriately cited throughout.

I am glad I could contribute to improving your this MS and hope my comments will be helpful. Please find my detailed point-by-point feedback attached below.

Comments for authors:

L 39–40: Avoid “probably due to…” I suggest rephrasing.

L 50: I suggest deleting “that causes damage to numerous crops,” as you mention the number and types of hosts in subsequent lines.

L 55–56: Perhaps add a brief mention of the feeding apparatus (rostrum).

L 71: H. halys… (abberv.)

L 93–94: I would suggest rephrasing as follows: "(e.g., Mikac et al., 2016 on corn rootworm; Lemic et al., 2016 on sugar beet weevil; Benitez et al., 2014 on wireworms, among others)."

L 108–110: I suggest a small adjustment for improved flow: "In this study, we hypothesize that morphological variability within populations of H. halys in southern Europe, as determined through geometric morphometric analysis, contributes to the species' adaptability and invasion success in the region."

L 147–150: I suggest a small rephrasing for clarity: "Landmark coordinate data for anterior and posterior wings were processed using Generalized Procrustes Analysis (GPA) (Rohlf and Slice, 1990), which standardized the digitized individuals by removing variation in size, position, and orientation, aligning them based on centroid size."

L 154–157: A brief explanation of Mahalanobis distances would make this part more informative for readers unfamiliar with the concept.

L 217–220: I would propose the following rephrasing for more clarity: "Although the allometric effects were statistically significant, they did not substantially influence the overall shape variation observed, suggesting that other factors, such as genetics, may have a more prominent role in driving morphological differences."

L 237: If possible, could you define what is meant by 'slightly larger'?

L 261–264: The sentence is too long and complex. I suggest rephrasing as follows: "Wing shape and size reliably captured invasion-related patterns, aligning with findings from genetic studies in the same regions."

L 261–283: Some points, such as the comparison of genetic and morphometric results, are repeated multiple times. Consolidating these would make the discussion more concise.

L 331–345: After discussing the unique adaptation patterns of specific populations (e.g., Croatia's higher plasticity or Greece's lower plasticity), consider highlighting how these differences might influence pest control strategies to add additional impact to your manuscript. For instance, suggest that regions with higher plasticity may require more dynamic, adaptable control measures targeting multiple life stages or environmental contexts, while regions with lower plasticity may benefit from more focused, environment-specific interventions.

L 368–373: In the final paragraph discussing future pest management, add a note that phenotypic plasticity could guide the timing of interventions (e.g., early in adaptive cycles or after population establishment) and the selection of management tools that account for local ecological pressures.

L 374–378: The final paragraph could be strengthened to ensure clear conclusions. My suggestion:

"These findings emphasize the importance of phenotypic markers, such as wing morphology, for monitoring invasive species, particularly in scenarios where genetic methods are less feasible or cost-effective. The demonstrated ability of morphometric methods to reveal invasion-related patterns highlights their potential to complement existing approaches in integrated pest management. Expanding this research to include a broader geographical range and additional stages of invasion will further validate the utility of these methods. By integrating morphometric tools into monitoring and control programs, we can enhance the effectiveness of invasive species management strategies."

References:

L 402, missing doi = 10.1371/journal.pone.0149975.

Figures 5–9 included in the supplementary material are very blurry, and the titles are not readable. However, it is unclear whether this is due to the PDF document itself or if it will be corrected in the electronic version after editing by the publisher.

Kind regards,

Reviewer

6. PLOS authors have the option to publish the peer review history of their article (what does this mean?). If published, this will include your full peer review and any attached files.

Reviewer #1: **Yes: **Olivier Hendrik Berteloot

Reviewer #2: No

---

## [Author Response · Author response to Decision Letter 1]

23 Jan 2025

December 30, 2024

Dear Dr. Vazrick Nazari,

Please find below our detailed response to the reviewers’ comments.

We thank you, and the reviewers for their considered and helpful comments and look forward to working further with you to have our manuscript published in PLosOne.

Sincerely,

Darija Lemic

Response to reviewers:

Reviewer #1: Overall appreciation of the manuscript:

This journal article draft presents an interesting and relevant investigation into the morphological wing variations of Halyomorpha halys populations across southern Europe. While the study successfully explores the potential of wing geometric morphometrics as a cost-effective alternative to genetic methods, several areas could benefit from improvement.

The manuscript is somewhat unconcise in its results section, where results are interpreted, discussed and linked to speculations. Additionally, the manuscript the discussion is rather repetitive, particularly regarding the role of phenotypic markers, which can be streamlined to improve clarity.

Speculative statements, such as suggesting lower phenotypic plasticity in the Greek population without direct supporting evidence, weaken the overall argument and should be more cautiously framed. Additionally, the manuscript could do more to connect the morphological findings with genetic data, offering a clearer explanation of how these two aspects interact (and not interact). Some sections also require a more detailed explanation of the methodology and a stronger justification for the conclusions drawn, especially when discussing the potential role of genetic drift or adaptation. Overall, while the study contributes valuable insights, the draft would benefit from tighter organization, clearer and concise explanations, and more careful interpretation of the results.

Comments, questions and suggestions to be addressed:

Line 28-29: (suggestion for readability): analysing the size and shape morphology of the anterior and posterior wings

R1: Done

Line 32: Specify exact number of specimens and wings analysed

R2: Done

Line 52: “Harmful effects” are not ascribed to a pest but rather to a chemical, please rephrase this sentence

R3: Done

Line 55: Technically stink bugs like H. halys have a piercing-sucking mode of feeding, not plant juices are sucked, but more specifically pre-digested plant mass after injecting digestive enzymes into plant tissues. Please specify this better

R4: Done

Line 61: (suggestion for readability): Over the past three decades, it has spread rapidly through various pathways, including air transportation, cargo shipments, plant trade, and vehicles.

R5: Done.

Line 63: “significant spread” what is meant by this?

R6: We rewrite this sentence to improve readability.

Line 63: specify “the millions of dollars of damage” with a cited example, otherwise this is a vague statement

R7: Example added.

Line 66: Might be slightly over stated, not in Scandinavian and Baltic countries.

R8: Improved.

Line 68: (suggestion for readability): became => emerged

R9: Done

Line 70: Successful global invasion not only due to its ability to adapt but also (and mainly) its polyphagy, high mobility (active and passive)

R10: Sentence has been improved.

Line 71-73: threat and invasion rate causes are mixed here rephrase with Line 70 to clarify why it has had invasive success and poses a major threat.

R11: Sentence has been improved.

Line 74: Vague statement and a missed opportunity to connect genetic variability directly to the phenotypic observations or invasion dynamics

R12: The sentence has been reorganized to emphasize the connection between genetic variability and phenotypic traits.

Suggestion:

In the context of invasive species like Halyomorpha halys, genetic variability plays a crucial role in enabling populations to adapt to diverse environments and expand into new habitats, ultimately driving their invasion success (Gloss et al., 2016)

R13: Added into manuscript.

Line 75: These studies using COI and COII sequences should be cited at the end of this sentence.

R14: References added.

Line 88: Are there more European studies on this?

R15: More studies from Europe added.

Line 90: Statement might be overly strong, genetic diversity studies & morphological variability studies would be complimentary

R16: Sentence has been improved.

Line 91-94: “numerous studies” and “many others can come across as vague. Replace with more precise statements that conveys specificity and credibility

R17: Suggestion accepted.

Suggestion:

Previous research has shown that morphological differences can provide valuable insights into pest adaptations to agroecological conditions and invasion pathways, as demonstrated in studies on the corn rootworm (Mikac et al., 2016) and sugar beet weevil (Lemic et al., 2016).

Line 94-96: “may reveal”

R18: done

Line 101: Here the statement of complimenting genetic marker studies is made, I refer back to the comment in Line 90. Rephrase this in one of the statements and avoid repetition.

R19: The paragraph has been reorganized to avoid repetition.

Line 103-105: Too many unnecessary words like ‘indicating a research gap that needs to be filled”

R20: Suggestion accepted.

Suggestion:

Despite its recent establishment in Europe, the morphological variation of H. halys in these regions remains poorly understood, highlighting the need for research to elucidate its invasion patterns.

Line 105-107: Also lacks conciseness, clarity and academic tone

R21: This sentence has been removed.

Line 108-118: The hypothesis and objectives are restated in slightly different words, making the section unnecessarily repetitive.

R22: This section has been removed.

Line 127-129: repeating "multi-country, multi-year" can detract from an academic tone

R23: Removed.

Line 131: When discussing the totality of samples use N = 540 instead of n = 540

R24: done

Line 142: Sentence too long for readability it should start a new sentence after “camera”. Start new sentence with: The images were saved in JPG format…

R25: Done

Line 148: Unnecessary use of semicolon: connect sentences with a word or split sentence.

R26: Done

Line 152-154: Double use of angle brackets can be avoided by phrasing it like this for instance:

R27: Suggestion accepted.

To visually examine shape variation among countries, a Canonical Variate Analysis (CVA), a type of discriminant analysis designed for more than two groups, was conducted.

Line 154-157: The phrase “using permutation test” is missing the article “a” before “permutation test,”. It should be corrected to “using a permutation test”. An additional suggestion to clarify the text could be:

Mahalanobis distances were calculated to compare morphological differences among the four countries and were statistically tested for significance using a permutation test with 1,000 iterations.

R28: Suggestion accepted.

Line 159: A comma is needed after “Additionally”

R29: Done

Line 165: then => subsequently

R30: Done

Line 166-167: Correct in-text citation for R is (R Core Team, 2024). Additionally, provide a more concise phrase to mention that the statistical analyses were done in R.

R31: Corrected citation was included, and an additional text was included as the reviewer requested.

Suggestion:

All statistical analyses were done in R v.4.1.1 (R Core Team, 2024) using packages x and y

Line 174 -175: The explanatory power of the principal components is not consistently rounded (78.00% and 46.00% would then be correct). In this case one decimal place seems enough.

R32: Corrected.

Line 178-183: This section reads more like an interpretation of the results and could better fit in the discussion. Language like “highest variability” and “tended to be wider” is interpretation of the results and needs to be moved to the discussion or results need to be more concisely given such as the covariance matrix and the quantitative variability metrics in figure 3.

R33: Sentence has been improved.

Line 185-191: The results lack quantitative data (e.g. Mean wing width, variability metrics) and are given in an overly descriptive way rather than presenting the data concisely and leaving interpretations for the discussion.

R34: Geometric morphometrics data is usually explained in this way, the way the shape change is descriptive of the shape, we did not use mean wing, width, because it is not the aim of GM data results, this kind of "quantitative data" is shown in terms of ANOVA or permutation... this mostly explain the shape changes of the figure

Line 193-202:

1. The sentence "The CVA clearly showed that the Croatian population (represented by red dots) differed the most from the other populations" repeats information already implied in the subsequent text about Croatia being distinct.

R35: the repetitive text “with Croatia standing out as the most distinct group” was deleted

2. Mentioning PCA in a CVA-focused section creates confusion

R36: Fixed and data extra of PCA was removed to not create any confusion to the reader

3. The narrative includes subjective terms like "clearly showed" and "stood out," which should be avoided in the results section to maintain objectivity.

R37: Suggestion accepte.

4. The phrase "showed less variation between groups" is vague without quantitative or statistical evidence.

R38: we modify the sentence more related to what the PCA shows, the Slovenian population showed less dispersion between individuals

5. Statements such as "These results reflect considerable morphological diversity" and "Croatia standing out as the most distinct group" move toward interpretation, which is more appropriate for the discussion section.

R39: Suggestion accepted, and terms like this have been removed.

Line 204-211:

While mentioned in Table 2, this section does not provide numerical values (e.g., Mahalanobis distances, percentage variance explained by the CVs) to substantiate claims of differentiation or overlap.

R40: No needed to be included… the CVA is a discriminant analyses the % of variance is not real the only % of variance which should be shown in GM is the PCA which is the real variation, the CVA is an analysis which help to discriminant graphically a data… and work on that stand graphically showed in the plot.

Line 213-220

1. "small but statistically significant" could be more precise. The magnitude of 4.72% and 10.8% already conveys the effect size.

R41: Indeed, a small % of allometry could affect the shape of a population is the reason that in geometric morphometrics even if the % is small but significant should be reported.

2. The sentence "Despite these statistically significant results, the allometric effects were not sufficient to significantly alter the overall shape variation observed" is repetitive, as the lack of a dominant allometric influence is already implied.

R42:

3. While p-values are reported, the biological relevance of 4.72% and 10.8% of variation explained could be contextualized further.

R43: Allometry is the effect of size on the shape variation, the explanation in the text is already included… I think extra information could be repetitive size could be a determinant of shape variation but normally this is when allometry is bigger

Line 222-228:

The statement "the primary variation is driven by factors other than size alone" is speculative without direct evidence from the study.

R44:”a may be” was included in the text to not imply the sentence directly and keep the conditionality

The sentence "this effect was not significant enough to alter the overall pattern of shape variation observed in the population" repeats information already implied by the percentages and p-values.

Suggestion for concise results reporting (applicable for previous and next result sections as well):

In male wings, allometry accounted for 4.5% of the shape variation in the anterior wings (p < 0.001). In the posterior wings, allometric effects were slightly more pronounced than in females, explaining 13.8% of the shape variation (p < 0.001). While statistically significant, these effects were modest and insufficient to significantly influence the overall shape variation patterns observed in the population (Figure 8).

R45: Suggestion accepted.

Line 230-237:

1. The description of centroid size distribution is qualitative (e.g., "relatively conservative", "greater variation") without providing actual metrics or statistical support.

R46: Is a geometric morphometric study, this is shape, when a Size is used could be a data “quantitative” the centroid size is the size of the shape…

2. The term "relatively conservative" is vague and could be replaced with a more specific description of the size distribution.

R47: The sentence has been rewritten.

Line 238-241:

1. "the wing size range of wing size" is repetitive and unnecessary.

R48: Corrected

2. Terms like "more homogeneous" and "moderately high" are unclear without specific quantitative metrics or context.

R48: Its usual way to explain a more visual graphics in geometric morphometric, not needed to add quantitative data here.

Line 243-253:

1. Phrases like "indicating that Greece may be more genetically isolated" and "possibly indicating that they originated from a similar source" are speculative and not directly supported by the study's data.

R49: Paragraph has been rearranged and rewritten.

2. The phrase "a mix variation from the last two" is vague and unclear

R50: Paragraph has been rearranged and rewritten.

Line 262: "reliably capture the patterns associated with the invasion process" is broad and could be more specific, highlighting exactly what patterns were observed (e.g., differentiation, variability).

R51: Sentence has been improved to bee more precise.

Line 264: The parenthetical "see: Gariepy et al., 2021; Yan et al., 2021, Cesari et al., 2018" is conversational and less formal. Standard citation style should be used.

R52: Corrected.

Line 266: "early to complete establishment phase" is unclear and could benefit from more precise language.

R53: Corrected.

Line 268: "Over 2,000 wings analysed" could mention the exact number for precision.

R54: Done

Line 268: "Represents one of the most comprehensive" is generic and lacks detail.

R55: Corrected.

Line 271: The mitochondrial genes COI and COII should have been mentioned earlier in the introduction, and would then not necessarily have to be explained in the discussion.

R56: Suggestion accepted.

Line 277: Citation style is informal ("see Musolin et al.").

R57: Corrected.

Line 278-279: suggested revision:

These findings align with genetic marker studies (Gariepy et al., 2021; Yan et al., 2021), which demonstrated comparable population differentiation.

R58: Suggestion accepted.

Line 281: direction => pattern

R59: Done

Line 282: parenthetical reference to Fig. 7 should be (Fig. 7)

R60: Done

Line 282: Start a new sentence and clarify that this was from genetic evidence: “Greek population was shown to originate from multiple sources”.

R61: Sentence has been improved.

Line 285: "Most likely originates" is overly speculative and could be made more precise with supporting evidence. “likely originated from neighbouring Italy”

R62: Corrected.

Line 288: Specify that the interpopulation variation is in the morphological diversity

R63: Corrected.

Line 293: (as discussed in Musoulin et al., 2018) does not need parentheses and the citation contains a typo and is likely: Musolin et. al., 2018

R64: Corrected

Line 296: The explanation that "small sample sizes" account for the discrepancy is plausible but insufficient. Small sample sizes might affect genetic diversity estimates, but their connection to higher observed morphological variation is not explicitly explained.

There is no discussion of how genetic diversity (or lack thereof) in mitochondrial markers COI and COII is mechanistically linked to phenotypic diversity in wing morphology. Mitochondrial genes primarily affect metabolic functions, so their direct influence on wing shape variability is unclear.

R65: Paragraph has been added into manuscript.

Line 297-299: The statement "too small sample size remains a problem" is overly general. It does not specify what aspect of Xu et al.'s findings is affected by small sample size (e.g., underestimated genetic di

---

## [Editor Report · Decision Letter 1]

29 Jan 2025

Tracing the Invasion: Wing Morphometrics Reveal Population Spread and Adaptation Patterns of Halyomorpha halys (Stål, 1855) Across Southern Europe

PONE-D-24-53394R1

Dear Dr. Lemic,

We’re pleased to inform you that your manuscript has been judged scientifically suitable for publication and will be formally accepted for publication once it meets all outstanding technical requirements.

Kind regards,

Vazrick Nazari, PhD

Academic Editor

PLOS ONE

---

## [Editor Report · Acceptance letter]

PONE-D-24-53394R1

PLOS ONE

Dear Dr. Lemic,

I'm pleased to inform you that your manuscript has been deemed suitable for publication in PLOS ONE. Congratulations! Your manuscript is now being handed over to our production team.

Kind regards,

on behalf of

Dr. Vazrick Nazari

Academic Editor

PLOS ONE